# Accuracy of Solid-State Residential Water Meters under Intermittent Flow Conditions

**DOI:** 10.3390/s20185339

**Published:** 2020-09-17

**Authors:** Francisco J. Arregui, Laura Pastor-Jabaloyes, Angel V. Mercedes, Francesc J. Gavara

**Affiliations:** 1ITA-Grupo de Ingeniería y Tecnología del Agua, Departamento de Ingeniería del Agua y Medio Ambiente, Universitat Politècnica de València, Camino de Vera s/n, 46022 València, Spain; laupasja@ita.upv.es (L.P.-J.); angelmercedesg@gmail.com (A.V.M.); 2Head of Metering Department, FACSA, Calle Mayor 82-84, 12001 Castelló de la Plana, Spain; fjgavara@facsa.com

**Keywords:** solid-state water meters, static meters, electromagnetic water meters, ultrasonic water meters, water meter accuracy, intermittent flow, water meter errors

## Abstract

Accurate water consumption measurement of customers is a crucial component of water utility sustainability. During the last decade, sophisticated measuring technologies without moving components, known as solid-state water meters or static meters, have emerged. Solid-state water meters promise an improved accuracy with more processing and transmission capabilities in comparison with traditional mechanical meters. A compromise needs to be reached between energy consumption and battery life as all these new features are extremely demanding on electric energy. The usual approach adopted by the manufacturer is to reduce the frequency with which static meters take measurements of the circulating flow. This reduction in signal sampling frequency can have a significant effect on the accuracy of the instruments when measuring water consumption events of 30 s or less, these events being common in residential customers. The research presented analyses of the metrological performance of 28 commercially available solid-state water meters from six different manufacturers in the presence of intermittent flows of various durations. The results show that the magnitude and dispersion of the error under intermittent flows is significantly larger in comparison to steady state flow conditions. The ultrasonic meters examined were more influenced by the intermittency than the electromagnetic meters.

## 1. Introduction

Solid-state water meters or static meters are measuring instruments which do not use a mechanical measuring principle to quantify the amount of water consumed. Compared to mechanical water meters, static water meters have no moving components subject to wear. Flow rate and volume calculations use static sensors as the physical principles of measurements. In other words, flow rate figures are determined indirectly from the measurement of physical magnitudes, like the time difference of two sound waves travelling in different directions or the voltage between two electrodes. Most meters, commonly known as smart meters, which can analyze and transmit water consumption information, are static meters. However, it is possible to find in the market mechanical meters with an electronic register that can conduct the same type of analysis as a static smart meter with the only difference that the measuring principle is mechanical. Examples of these are velocity-single-jet, multi-jet, and Woltmann meters, or positive displacement meters—oscillating piston and nutating disc meters—equipped with an electronic register. This configuration is available from most of the leading water meter manufacturers around the world.

The absence of moving parts presents significant advantages over the traditional mechanical water meters. The first, and most obvious one, is that the unavoidable wear and tear of all moving components associated with mechanical technologies does not affect static water meters [1,2]. A static meter can work for long periods at high flows without any critical damage. Consequently, a slightly undersized static meter does not become a severe problem. It may be a suitable option for measuring water consumption of large customers having a high monthly consumption rate. On the other side, the degradation of mechanical components has a clear effect on the low flow sensitivity and their ability to measure internal leaks [3,4,5,6]. Mechanical meters tend to lose their performance at low flows rapidly and the starting flow rate gradually increases as the meter ages [7,8,9,10]. 

Furthermore, the achievable low flow sensitivity, even when newly installed, is significantly better in a static meter in comparison with a standard velocity mechanical meter and comparable to the best positive displacement meter. It is not unusual to find in the market static meters with a metrological classification of R400 or better according to ISO 4064-1:2014 [11]. Finally, it is essential to mention that static meters can achieve and maintain their excellent metrology at any position of installation. Contrary to what usually happens with mechanical meters, the metrological performance is not affected by the orientation (vertical or inclined) of the meter [12].

Another major advantage of static meters over mechanical meters is their insensitivity to poor water quality. Static meters are typically more stable against limescale deposits and loaded waters. Nevertheless, depending on the construction, static meters are not utterly unaffected by water quality effects. In some cases, extremely loaded, fouling, or abrasive waters can damage or disable the sensors or obstruct the measuring tube with similar consequences as when a turbine of a mechanical meter is blocked or damaged [13].

However, there are also several disadvantages of using static meters over mechanical meters. The actual durability in the field of the batteries, under real working conditions, is still unknown as many of these meters have only been installed by water utilities for the last few years. Moreover, the use of Lithium batteries has a considerable negative effect on the environment, and the recycling cost of the instruments, including the battery, should not be neglected.

Additionally, at very high flows, the metrological performance of static meters can also be unreliable. While mechanical meters show problems at high flows related to the magnetic coupling between the turbine and the register, the algorithms of some static meters struggle under the occurrence of high flows, larger than the overload flow, Q_4_. Most ultrasonic meters have a high-flow cut-off at which the meter saturates or even stops counting.

However, the main drawback of static measuring techniques is associated with the requirements of the electric supply of the sensors used to measure water flows and all electronic components needed to perform water consumption calculations and data transmissions. This means that an expensive large capacity Lithium battery is required to power the meter during its expected useful life, significantly increasing the cost of the device. Considering that the typical, expected useful life of a residential meter can be in the range between 8 and 15 years, all efforts need to be put in the design of the meters to use as little power from the battery as possible. For this reason, static meters are not continuously measuring the flow or powering the electronic components needed to perform the calculations. To extend the battery life, static meters sample the flow signal at periodic intervals. The periodicity depends on the type of meter and its design but, in most cases, is in the order of 5–6 s, although some models can lower this period down to 1 s or less. This means that there is a considerable chance that short water consumption events are not properly measured.

This limitation can become a problem when measuring the water consumption of residential customers. In fact, a substantial amount of water consumption inside houses, 20% or more depending on the type of househould, has a duration of less than 30 s [14,15,16,17,18]. Short duration events can have a very negative effect on the overall metrological performance of the meters, depending on how the consumption and the sampling of the flow signal are synchronized. Briefly, and in order to show the impact of sampling on the accuracy of a water meter, consider a water consumption event having a duration of 29.8 s. This consumption is measured by a static meter with a sampling interval of 6 s. For this event, the worst case scenario is when the flow is only measured at the following instants: 5.9, 11.9, 17.9, and 23.9 s. Simplifying the calculations, for the static meter, under these extreme conditions, the duration of the consumption is only 24 s instead of the actual 29.8 s. This means that the measuring error would be close to −19.5%. Obviously, this is an extreme situation, but it does not account for other effects like the time needed by the water meter to conduct the calculation of the circulating flow rate and the influence of unsteady flows that occur at the beginning and end of the consumption. If the duration of the water use events decreases, the measuring error increases. With the purpose of minimizing the problem, a frequent informal argument given by most static water meter manufacturers is that measuring errors caused by signal sampling will eventually compensate each other. However, current water meter standards, including the latest version of the ISO 4064 published in 2014 [19], do not include specific tests that verify that the calculation algorithms used by the water meters compensate sampling errors in the medium-long term. In other words, at the present time, the algorithms used must be tested to verify that they do not introduce a bias that could benefit either party, the customers or the water utilities.

Up to now, and due to the novelty of these metering technologies, there are very few bibliographic references on the metrological performance of water meters under intermittent flows or short duration water consumption events. The most recent one was by J. R. Chadwick [20] in which the accuracy of residential water meters in response to burst flows (of 1 s or less) was investigated. In this research, two of the meter types tested were ultrasonic. A specific mention on the influence of the sampling rate was made. Unfortunately, this research only considered very short consumption events of 1 s or less, which are not usual in households and are not representative to establish the overall measuring performance of static meters. L. Hovany [21] and S. Yaniv [22] analyzed the performance of several mechanical meters under the pulsating flows caused by a device designed to reduce unmeasured water caused by leaks inside homes. The research only considered mechanical meters.

The analysis conducted focuses on the metrological performance under an intermittent flow of small size (DN15 and DN20) static water meters typically used to measure residential water consumption. The measuring technologies considered were ultrasonic (transit time) and electromagnetic. However, for comparison purposes, mechanical velocity meters were also added to the test sample. In total, 10 different meter types produced by seven manufacturers were considered. The main objective of the research was to establish the influence of intermittent flow conditions of various durations on the measuring errors. To obtain realistic figures, closer to the ones that would have been reached in the field, all water meters were tested without activating the test mode. This mode increases the sampling frequency and reduces the reading scale interval.

For all meters, the metrological performance obtained under steady state flow conditions, as defined by ISO 4064-1:2014 [11], was compared with the measuring errors found under intermittent flows. Each one of the test conditions, defined by the flow rate and the intermittency period, were repeated up to a maximum of 17 times and a minimum of three times to establish the error distribution and to allow for statistical comparison methods to be applied. The statistical analysis was conducted using R-statistics [23]. The external package needed in each case is detailed in the corresponding section.

## 2. Materials and Methods

The test programme was designed to understand the actual water meter performance working under similar operating conditions and with the same configuration as in the field. For this reason, and with the purpose of identifying significant changes in the metrological performance of the meters, all tests had to be carried out, ensuring that the experimental uncertainty was below an acceptable threshold. Under this hypothesis, it is important to realize that the greatest contribution to the test uncertainty is caused by the resolution of the water meter reading. This resolution for a typical domestic water meter is 1 L. This value does not meet the published standard requirements. As stated by the ISO 4064-1:2014 [11] and the OIML R49-1:2013 [24], the subdivisions of the reading scale of water meters having an accuracy class 2, should allow for an error of estimation due to water meter resolution of less than 0.5% of the volume corresponding to 90 min at minimum flow (Q_1_). However, the ISO 4064-1:2014 [11] and the OIML R49-1:2013 [24] also allow for ancillary devices to improve the reading resolution of the water meters. The use of these devices or any button or system that enables an improved volume resolution in the display, in order to meet the requirements defined in the standards, may “inform” the water meter that is being subject to a test, modifying the internal operating conditions. In other words, there is no means of guaranteeing that the internal algorithms or operation of the water meter do not change when pressing a button with the theoretical purpose of improving the resolution of the meter. For this reason, during this work, water meters were tested in the laboratory without activating the test mode (common in these electronic meters) or pressing any button that increased the volume resolution of the display.

### 2.1. Test Bench Description

Two different volumetric test benches were used during the experiments. The first one was used to carry out the tests under steady flow conditions. The second one, a built-to-purpose test bench, was employed for the tests conducted under intermittent flow conditions.

A simplified schematic of the first test bench is shown in Figure 1. Water is pumped from an underground tank using variable speed pumps. The stability of flow and pressure at the inlet of the bench is assured by means of a 1000 L pressure vessel. The bench can fit up to five DN15 water meters in series. Downstream the bench, there is a set of valves and flow meters that allow adjusting the flow rate of the test to the desired value. Two probes of 10 and 200 L are used as a reference volume. For the series of tests presented in this study, the 10 L probe was only used at low flows, i.e., 20 and 50 L/h. For the remaining tests, the 200 L probe was used in all cases. The errors of the meters were obtained by means of the standing start and stop test method (ISO 4064-2:2014 [19]). The scale division of the 10 and 200 L probes was 0.01 and 0.2 L, respectively, which represents 0.1% of the tested volume and is significantly smaller than the volume resolution that can be read from the meter.

The expanded uncertainty of the tests conducted in this work was mainly driven by the volume resolution of the meters. For all digital meters under examination, the smallest volume that could be read from the register without interfering with the instrument was 1 L. Considering that the volume indication of the meter has a discontinuous movement, the error due to the resolution of the meter can be estimated, according to ISO 4064-2:2014 [19], as 1% or less when the 200 L probe is used (a graphical example is presented in Appendix A). The uncertainty originated from the scale division of the 200 L probe, considering that it has a continuous movement, and only one reading is necessary to estimate the volume, can be estimated as 0.05%.

The second test bench (Figure 2) has two parallel lines, which allow doubling the number of meters, which can be analyzed simultaneously. An electronic controller opens and closes the shut-off electrovalves of each line at the pre-set times, depending on the type of test to be conducted. This configuration made it possible to have one line running while the other was closed, thus reducing the time needed for the experiment.

This test bench uses three brand-new DN15 positive displacement meters as master meters for reference. These meters, known to be extremely repetitive, were tested against the 200 L volumetric probe to obtain a detailed error curve at different flows (Figure 3) and to verify that their measuring errors did not change significantly under intermittent flow conditions. One meter is installed at the downstream extreme of each parallel line, and the third meter is located at the downstream end of the bench. This configuration allowed for redundancy in the measurement of the total volume passed through the meters. Once the volumes are corrected with the error of the meters, the sum of the measures taken by the two upstream meters must be equal to the total volume measured by the downstream meter.

The set of regulating valves installed downstream the two parallel lines allow for the adjustment of the magnitude of the flow rate passing through the meters. By activating and deactivating the different branches of this set of valves, it is possible to select the flow rate. The combination of electrovalves allowed not only to produce intermittent flows but also to modify the magnitude of the flow during each activation period. This way, the experiment could be set to operate cyclically, which helps other authors to reproduce the tests more easily using the same operating conditions. The cyclic operation also simplifies the interpretation of the results. This type of testing is also a suitable procedure for checking the repeatability of the meters under study, analyze any potential bias in the internal measuring algorithms of the meters, and to decrease the influence of meter resolution. In addition, the simplicity of the tests helps to identify the operating conditions that most affect the metrological performance.

### 2.2. Sample Description

The purpose of the experiment was to explore the stability of measuring errors of small diameter solid-state water meters under the presence of unsteady and varying flows. Performance under these working conditions is critical for water utilities as varying consumption flow rates are typical of domestic customers and currently are not considered in any test programme defined in international standards related to water meters. Moreover, these standards only define accuracy tests under steady flows, limiting the maximum variability of flow during the tests to 2.5% if the flow is between Q_1_ and Q_2_ (exclusive), and 5% if it is between Q_2_ (inclusive) and Q_4_ (ISO 4064-2:2014 [19]).

For the study, a sample of different DN15 and DN20 meters available in the market from various manufacturers was provided by FACSA. In total, 35 m units were subject to test. The characteristics of the meters under examination, including technology, diameter, metrological class, and permanent flow, are presented in Table 1. This table provides an overall view of how the different types of meters are distributed.

The number of meters from each manufacturer is relatively small. The reason is based on two assumptions:(i)Manufacturers of solid-state water meters ensure minimal tolerances during production. In addition, variations detected between units are later corrected in the calibration process. Consequently, potential differences in the behavior of a solid-state water meter under steady and intermittent flow conditions are mainly due to the firmware and/or the signal processing algorithm, which are identical for all meters of a certain type and manufacturer.(ii)The present study aims to detect whether the processing algorithms used by each meter type show any significant fault that impedes a correct measure of water consumption under intermittent flow conditions.

Therefore, under these assumptions, a large sample of each type of meter was not necessary, since all units should perform in a similar manner. For this reason, the study, with the limited resources available, focused on testing meters from different manufacturers rather than testing several units of the same type.

Furthermore, solid-state water meters are a relatively novel technology. Manufacturers are extremely active, and brand-new meter types are presented continuously on the market. Additionally, like any other high-tech instruments, manufacturers constantly improve hardware and software to add new features. This becomes a significant issue as the metrological performance of the meters can be greatly affected by the firmware of the instrument. Therefore, it is not possible to guarantee that the behavior of a meter manufactured today will be the same as one produced in a few years, even if they share the same exact external appearance and measuring features.

For this reason, the main purpose was not only to provide an example of how solid-state meters available today could perform in the field but also to highlight the importance of designing a new test programme that needs to be included in the water meter standards. These new tests should analyze the metrological performance of the meters under operating conditions more similar to what meters will find in the field. Additionally, for comparison purposes, a batch of a widely used single-jet mechanical water meters, M6 and M9 type in Table 1, was added to the sample under analysis.

For reference, Table 2 shows for each type of meter the average age and accumulated volume, the default reading resolution, and the availability of a test mode (resolution change). Except for M2 and M6 meter types, all water meters tested were out-of-the-box brand new meters. In the particular case of M2, some of the meters tested were manufactured in 2014, while others were produced in 2017 and 2018. All these M2 meters have been in operation and removed from the field except the units named M0007 and M0009. Meter type M5, of which only one unit was available, did not properly function at the beginning of the tests. The display of the meter indicated that water was passing through it, but the index of the register did not show any increment in volume. Unfortunately, it was not possible to replace the unit with an operational one. However, it is noteworthy that a solid-state meter, being in apparent good working conditions, did not sense any flow. This meter could have been installed in the field without noticing it was defective. In any case, and in order not to distort the results, this unit was not included in the statistical analyses conducted.

### 2.3. Test Programme Description

The test programme started with a series of experiments designed with the purpose of obtaining the reference error curve of each meter so it could be used for comparison with other tests. This curve was obtained under steady state conditions as defined by ISO 4064-2:2014 [19] employing the test bench described in Figure 1. Except at low flows, the minimum volume passed through the meter in each test was, at least, 200 L. For this reason, the uncertainty in estimating the error of indication is mainly driven by the minimum resolution with which the accumulated volume of the meters can be read. Overall, this figure is close to 1%.

In order to improve the reliability of the reference error curve, and to assess the repeatability of the meters, each flow rate was tested several times. Due to restrictions in the duration of the tests, the number of repetitions did vary from 17 at high flows down to 3 at lower flows. Additionally, not all meters were tested at the same flow rates, and some of them, received after the test programme was already started, were tested at a reduced number of flow rates (those strictly needed for comparison with the unsteady flow conditions). More information about the number of repetitions conducted by meter and test type is provided in Appendix A.

The details of these tests, used as a reference, named T1, are shown in Figure 4. They were conducted under steady state conditions (regime: S), the flow rate through the meter was maintained constant (flow variability: C), and the flow rate ranged between 20 and 5000 L/h. Considering that the flow rate was kept constant during the test, the Cyclic Period parameter does not apply in this case (N/A). The numbers in brackets indicate the quantity of tests included in the group. In total, 334 tests were conducted on T1.

Additionally, to the above, the study comprises a set of tests conducted under intermittent flow conditions to verify the ability of the meters to measure short consumptions and to adapt to real working conditions in the field. With the purpose of setting up a reproducible experiment, the configuration of these tests only modifies two parameters: the cyclic periodicity and the flow rate. By doing this, it was possible to create consumptions of a specified duration and flow rate that were repeated over time (Figure 5). For example, the group named T2 (Figure 4) corresponds to tests in which the duration of the consumption was 2 s ($\Delta t_{t}$ = 2 s), and the flow rates were 200, 500, and 2000 L/h. The tests in the group T3 were conducted at the same flow rates, but the cyclic periodicity was changed to 5 s ($\Delta t_{t}$ = 5 s), simulating consumptions of a longer duration. In a similar manner, the tests in groups T4 and T5 were performed with the cyclic periodicity set to 10 and 20 s, respectively.

To add more controlled variability to the operating conditions during the assays, the tests named T6 and T7 considered intermittency and flow rate variability within the same experiment. More precisely, three flow rate profiles were established: (a) flow profile P1 covers the lower flow rate range, alternating during the activation period 600 ($Q_{t}^{1}$), 400 ($Q_{t}^{2}$), and 200 L/h ($Q_{t}^{3}$); (b) flow profile P2 takes into account higher variability and a larger average flow rate than P1, with, $Q_{t}^{1}$, $Q_{t}^{2}$ and $Q_{t}^{3}$ set to 1100, 900, and 200 L/h, respectively; (c) finally, the flow profile P3 only covers the upper flow rate range for which the values of, $Q_{t}^{1}$, $Q_{t}^{2}$ and $Q_{t}^{3}$ were set to 2000, 1500, and 1500 L/h. T6 uses a 5 s cyclic periodicity and was carried out with P1 and P3 profiles (see Figure 4). Test T7 was conducted with a periodicity of 10 s instead of 5 s, using all the previous flow profiles, i.e., P1, P2, and P3.

All tests conducted under intermittent flow conditions, from T2 to T7, were performed in the test bench described in Figure 2. For these tests, the required minimum duration or volume passed through the meters was 1 h or 200 L, whichever was more restrictive (Table 3). Consequently, for flow rates lower than 400 L/h the constraining condition was the minimum volume. On the contrary, the required condition for higher flows was the minimum duration of the test. These impositions reduced the uncertainty due to the volume reading resolution of the solid-state meters, which was 1 L in all cases, to a value lower than 1% (Table 3). The restrictions used also ensured that the number of flow activations during a test is sufficient to identify biases or defects of the algorithms and that these defects will have a clear impact on the measuring error.

### 2.4. Analysis Methods Overview

The available sample size does not allow for inferential statistical analysis to compare the behavior of meters between different technologies and models or against steady and intermittent flow conditions. For this reason, the analysis of the results is essentially descriptive. Furthermore, the repetitions of a certain test conducted on a meter were handled as individual data, instead of taking the average of the errors obtained as a representative value. In this way, the dispersion which is one of the most relevant characteristics under study, can be examined more clearly.

Thus, the analysis conducted was based on position and dispersion measurements to describe the results, as well as the use of graphic tools such as box-whiskers and histograms. On the other hand, the variables used to classify the results obtained into different groups were the flow rate, type of test, diameter, technology, type of meter, and flow conditions. The primary tool used to conduct the analysis described was R-statistics [23], and the package ggplot2 [25] was employed to generate plots. 

Despite of the above, the sample tested includes a number of M2 meters sufficiently large to apply the tools of inferential statistics: 13 units are available, five of them manufactured in 2014, and eight in the period 2017–2018. Hypothesis testing by means of parametric tests is the proposed statistical tool to compare the metrological performance of M2 meters: (1) of different ages; (2) under intermittent and steady flow conditions. Since the results obtained at this stage are preliminary, and the assumptions for this type of analysis must be verified by means of a larger sample, the methodology followed, and the results were included in in the Appendix A, respectively.

## 3. Results and Discussion

The results obtained from the tests underwent a preliminary validation process to identify outliers and abnormal data points. This validation process includes the identification of transcription errors of the meter readings by means of pictures taken at the beginning and end of each test and the consistency of the data. This consistency checks verified the volume recorded by the different reference meters or probes and the relationship between the duration and flow rate of the test and the volume used for reference. For example, in the case of the results obtained in the tests under intermittent flow conditions, the following was verified: (1) that the sum of the volume recorded by the reference meters of each test line is within ± 0.5% of the volume recorded by the reference meter at the bench discharge (results of this analysis is presented in Appendix A); (2) that the volume passed through each line is 50% ± 5% of the total volume recorded by the reference meter at the bench discharge (Appendix A). The volumes measured by these reference meters were corrected according to the corresponding error curve presented in Figure 3. Finally, in order to facilitate the statistical analysis conducted in this work, all valid data were organised into a relational database.

### 3.1. Metrological Performance under Steady Flow Conditions

The steady state flow tests were used for two main purposes. On the one hand, it was necessary to verify that the meters under analysis met the metrological requirements for new meters defined under the ISO 4064-1:2014 [11]. On the other hand, it was necessary to obtain a reference error curve that could be used for comparison purposes with other tests performed. 

Concerning to the first objective, Figure 6 and Figure 7 show the detailed results of the error tests per meter through boxplots. These graphs present the error distribution of the various repetitions conducted at each flow rate under steady flow conditions. Appendix A offers numerical details about the average error and the standard deviation of the tests. Like any traditional mechanical water meter, the error of the solid-state water meters under examination should be within the maximum permissible error of ±2% for flow rates greater than Q_2_ and an accuracy class 2.

Almost all meters tested met the ISO 4064-1:2014 [11] metrological requirements for the flow rates considered. M5 and M7 type meters were exceptions to the previous statement. As explained before, the only M5 unit available did not measure any flow, although the display of the meter and the meter itself seemed to be in proper working order. M7 meters showed an average error of approximately −8.5% at the highest flow rate tested of 5000 L/h, which for these meters corresponds to the overload flow rate (Q_4_). This malfunctioning at high flows has also been detected by the authors in other brands of ultrasonic meters and actual figures are detailed in the technical specifications of some brands. However, this behavior only appears under the presence of flow rates larger than 1.25·Q_4_ and disappears once the flow rate decreases below that threshold. Therefore, this malfunctioning is caused by limitations of the algorithms used to calculate the flow and not by a defective component of the meter. Nevertheless, these results at high flows confirm the importance of testing the meters over a wide range of flow rates before they are put into operation.

With regards to the single-jet mechanical water meters, M6 type has a slight tendency to over register water consumption as the flow increases (Figure 6). Meters units M0011 and M0012 exceeded the maximum permissible errors for flows greater than 1000 L/h. In addition, M0011 showed poor performance at a flow rate of 50 L/h, with an average error (five tests conducted) of −12.3%. This loss of accuracy is common in mechanical meters that have been in operation for several years (Table 2). In contrast, Figure 7 shows that the overall repeatability in a steady state test of a brand-new single jet mechanical water meter (M9 type) is satisfactory.

The steady state tests were employed not only to verify that the actual errors of the meters were within the maximum permissible errors allowed by the ISO 4064-1:2014 [11] standard but also to measure their repeatability. In this regard, the ISO standard establishes that the standard deviation of the errors at a given flow should not exceed one-third of the maximum permissible error, which for flows larger than Q_2_ is 2%. This means that the standard deviation of the errors should be smaller than 0.66%. However, provided that the reading resolution of the solid-state meters is limited to 1 L, the overall uncertainty of the tests is close to 1%. This means that even under steady flow conditions, a standard deviation well below the overall uncertainty cannot be expected from the tests. Considering this limitation and the results presented, it cannot be stated that the meters under analysis do not meet the repeatability requirements established in the ISO standard. Figure 8 consolidates the repeatability of the measuring errors obtained by technology, nominal diameter, and flow rate. Each box-whisker plot is built with the standard deviations that the associated meters have shown in each test. For example, the box-whisker plot corresponding to DN20 meters with EMF technology and a flow rate of 2000 L/h is composed of five data or, in other words, five standard deviations corresponding to meters M0021, M0032, M0033, M0034, and M0035 that were tested 7, 3, 3, 3, and 3 times (Appendix A), respectively, under steady flow conditions at 2000 L/h. The lower standard deviation achieved by mechanical meters can be explained by the better scale resolution of these meters.

For this reason, due to the poor volume resolution that is readable in the display of solid-state meters without interacting with them, it seems evident that the ISO standard needs a significant upgrade. Improving the scale resolution of the meters, available without any interaction with them, is the only option to conduct proper accuracy tests in a laboratory that ensure that meters will operate during the tests exactly as they will in the field. Currently, the ISO standard and the OIML recommendation specify that all meters should incorporate a verification device that “provides means for visual, nonambiguous verification testing and calibration”. The problem is that this requirement on the verification scale interval of the display of a water meter is only met when the test mode is activated. Consequently, this constraint related to the allowable resolution has not been properly interpreted by manufacturers, and currently, published standards do not require that the verification scale interval is permanently readable. This way, according to the ISO 4064-1:2014 [11] or the OIML R49-1:2013 [24], the required resolution of the verification scale of a meter having a Q_3_ of 2500 L/h, and a metrological class R160, is 0.0586 L. If the metrological class changes to R250 or R400, the required resolution decreases to 0.0375 and 0.0234 L, respectively. In all static meters tested, these resolutions can only be achieved by activating the test mode; otherwise, the scale resolution is 1 L. Unfortunately, once the test mode is activated there is no means of guaranteeing that the meter will have the exact same performance as with the test mode deactivated as this modes changes the sampling frequency and other operating conditions of the meter.

As expected, the error curve of solid-state water meters is relatively more uniform (flat) compared to the error curve of a single-jet mechanical water meter, which suffers from more oscillations throughout the measuring range. Figure 9 shows by means of a box-whisker plot, the error distribution at different flow rates of all units tested. Figure 9 does not include the results from tests at 50 L/h of the unit M0011 (M6 meter type) and at 5000 L/h of the units M0016 and M0017 (M7 meter type), with an associated error that exceeded ±2.5% of the reference volume, in order not to bias the results. An assessment of the variability of the error through the measuring range can be easily analyzed by the interquartile range amplitude of the box-whisker diagram.

Figure 9 also shows the ability of the manufacturers to produce meters with the same performance. Surprisingly, solid-state meters do not show any significant improvement in this respect to brand-new mechanical meters [26], especially when compared to oscillating piston meters. The tests conducted showed that meters of the same manufacturer can have an average measuring error throughout the tested range of more than 1.5%. Higher variabilities in performance indicate lower control over the production processes. This implies that water utilities need to implement stricter quality control procedures on the meters received from that manufacturer to guarantee that there are no defective units in the inspected lot. This is the case of M1 and M3 type meters. Figure 9 also shows that all solid-state meters, except M7 type at Q_4_, can easily maintain the errors within the maximum permissible error of 2% under steady state conditions.

The single DN15 electromagnetic meter unit under analysis, M4 type, showed a decline in repeatability at high flows (Figure 10). This behavior was not observed in the DN20 units by the same manufacturer, M10 meter type, which presented a more stable performance throughout the flow rate range. In any case, the repeatability of this technology during the steady state tests was better than ultrasonic meters.

The single-jet meters removed from the field, M6 type, presented the expected variability in the performance of meters that have been in operation during some time (Appendix A in Appendix A). Contrarily, the M9 meter type, which corresponds to a brand-new water meter, exhibits an extremely low variability of the error at each flow (thanks to a volume reading resolution of 0.1 L), with some oscillations throughout the flow rate range.

Figure 11 presents the consolidated variability of the errors obtained by technology and the flow rate. As expected, ultrasonic meters showed a uniform behavior throughout the measuring range, and the average error at each flow rate slightly oscillates around the average value. The error distribution of the electromagnetic meters, M4, and M10 types show a difference in behavior between the two, which essentially are the same meter of different diameters.

### 3.2. Metrological Performance under Intermittent Flow Conditions

Domestic water demand is exceptionally heterogeneous, and flow rate and duration of water consumption events are extremely scattered [14,15,16,17,18]. From a duration perspective, a shower event is not comparable with shorter uses, like filling a glass of water. Additionally, the flow rate of a leak is much lower than the consumption flow occurring when several water appliances are used simultaneously. Each water end-use has its own independent characteristics, which make the modelling of residential water demands a complex topic. However, some authors have proposed a simplification of all this casuistic by modelling water consumption events as a series of pulses of a given duration and a flow rate that are distributed through time, both of these parameters (duration and flow rate) being described by probabilistic functions that can be specific to a water end-use and individual user [15,27,28,29,30,31,32]. Therefore, from a standardization point of view, employing complex consumption profiles to conduct the tests is not an option, as the purpose of this experiment is to design and conduct a test programme that provides repeatable results and can be reproduced by an independent third party. The test programme used intended to limit the intermittency and variability of flow. In the case of flow intermittency, the cyclic periodicity of the consumption pulses was set to 2, 5, 10, and 20 s. In the case of flow rate, and provided that the typical consumption flow of a domestic appliance is between 200 and 2000 L/h, the test flow rates were primarily chosen in this interval.

As it has been already mentioned, sampling of the flow rate signal is a common technique to all solid-state meters to extend their battery life. The purpose of conducting metrological tests under intermittent flow conditions is to establish if the signal sampling has any effect in the measuring error of the meters. Some meters feature a fixed sampling rate, which typically is in the order of 5–7 s. Other meters are designed with a variable sampling frequency depending on the presence and magnitude of the flow rate. The algorithms that change the sampling frequency are confidential, and no details have been provided by the manufacturers.

Consequently, to obtain more realistic results, it was not acceptable to notify the meter that it was subject to test by interacting with it or activating the test mode. Therefore, as it has already been said, all meters were tested in the same conditions as they would have been in the field.

The analysis of the results of the experiments described in this section focuses on three main issues: (1) differences in metrological performance of the meters when subject to steady and intermittent flow conditions; (2) impact of cyclic periodicity and magnitude of intermittent flows in the measuring errors; (3) potential biases caused by intermittent flows that could favor one of the parties.

Figure 12 and Figure 13 describe through boxplots the error distribution of the tests performed under intermittent flow conditions at different flows by meter type. The error distribution obtained per meter can be found in Appendix A. It is important to highlight that these charts compile the raw results of all the test types under intermittent flow conditions defined in Section 2.3 or, in other words, the error obtained in each repetition of a test conducted, not the average error.

The results show that the error magnitude significantly increased when compared to steady state conditions. It was not unusual to obtain a measuring error of ±20% (the percentage of results in which the error of the meters tested was greater than ±5% for the various test types considered is detailed in Appendix A). This statement applies to a greater or lesser extent to all meter types. However, ultrasonic meters are more affected than the electromagnetic meters under examination. This is mainly due to the fact that signal sampling frequency is higher for electromagnetic meters (1 Hz or more) than for ultrasonic meters (0.2 Hz or less). Consequently, electromagnetic meters are more prepared for accurately measuring short duration consumptions events like the ones found in households.

Therefore, all meter types presented a significant difference in performance between steady and intermittent flow conditions. As will be shown later in the analysis, this difference is affected by the cyclic periodicity of the flow, the duration of a consumption event being more significant than the test flow. Although the results obtained must be contextualized, since the proposed tests magnify the potential biases associated with short consumptions events, it is a fact that the operating conditions in the field are continuously changing depending on the consumption profile. In the field, the error measuring a consumption event may be positive, and the following consumption event may be measured with a negative error. Therefore, the concern of water service managers is whether the errors of various signs that happen over time compensate each other in the long run. A more detailed analysis of the T6 and T7 tests (variable and intermittent flow conditions) was carried out in Appendix A to address this concern.

In the case of solid-state water meters, an increase in error dispersion was also observed. As explained in Section 2.3, the test volume of the tests under intermittent flow conditions is relatively large, and the expanded uncertainty of the test does not exceed 1%. Moreover, the errors obtained under steady flow conditions show a dispersion that meets the requirements of the ISO standard (1/3 of the maximum permissible error). Thus, the large dispersion of the errors found is strictly related to the internal algorithms and signal sampling periodicity. In addition, differences in performance between manufacturers and technologies can be easily identified, although a larger sample would have to be analyzed in the future to draw well founded conclusions. For example, the M7 and M2 meter types, both ultrasonic meters of different diameter (DN20 and DN15, respectively) from manufacturer B5, show more significant errors when the cyclic periodicity of the intermittent flow is set to 2 s (test type T2). Contrarily, the M1, M3, and M8 meters, which are also ultrasonic meters from three different manufacturers, have a more uniform response to different durations of intermittent consumption events. They even seem to achieve a worse metrological performance when the cyclic periodicity is set to 5 s. On the other hand, the electromagnetic meters tested show a better performance than the ultrasonic meters under all types of intermittency. Their average metrological response to the different intermittent flow tests is very similar to the one found under steady flow conditions. As explained above, this is mainly due to the fact that the signal sampling frequency is higher than 1 Hz. 

For comparison, the single-jet mechanical water meters in the sample are still remarkably repetitive, but there is a strong tendency to over register as the duration of the consumption event decreases. Nevertheless, this behavior tends to diminish as the flow rate increases. The over registration of the meter is caused by the rotational inertia of the impeller, which keeps turning for a period of time after the consumption has finished.

Despite the fact that the reference volume in each test was sufficient to reduce the uncertainty to less than 1%, a number of repetitions showed remarkably large errors. Table 4 describes the tests in which the errors obtained were above 50%. In such cases and considering that for steady state conditions, the errors of the meters were within the maximum permissible errors of the standard, it is considered that the meter presented an abnormal performance. Therefore, these results were excluded from further statistical analysis in order not to distort the boxplots and the conclusions obtained. It should be noted that the volume passed during the intermittent flow test is, as described in Section 2.3, equal to or larger than the volumes used in steady state tests. Hence, these results can only be interpreted as anomalous performances of the meters that need further investigation and more detailed analysis. In addition, it should be clarified that not all meters described in the sample could be tested under intermittent flow conditions. The tests and number of repetitions conducted on each meter are specified in Appendix A.

Figure 14 allows for a comparison between technologies in terms of repeatability and error bias, where the standard desviation and coefficient of skewness corresponding to error distribution under intermittent flow conditions are represented through histograms. It can be observed that errors of ultrasonic meters are more dispersed than for the electromagnetic units tested, especially meters belonging to M2 type. In turn, mechanical meters are by far the ones presenting the best repeatability, with the standard deviation that, in most cases, is less than 1%. Regarding the coefficient of skewness, it is symmetrically distributed around the zero value. Thus, the error distribution associated with a meter tested according to a type of test and a given flow or flow profile could show a positive skewness (i.e., the mean is greater than the median). However, the error distribution associated with another meter of the same type or even the same meter subject to another type of test could show a negative skewness. Therefore, this evidence that the internal algorithms of the static meters tested do not intentionally exploit the errors in any direction or cause a clear bias in the errors distribution.

To conclude this section, a detailed comparison of the results obtained under steady and intermittent flow conditions was conducted. To facilitate the analysis, tests results were grouped into two flow rate ranges: (1) the lower range comprises average flow rates between 200 and 500 L/h; (2) the upper range includes the errors obtained between 700 and 2000 L/h. Table 5 and Table 6 describe per meter unit the mean error and standard deviation associated with the tests performed under steady state (S) and intermittent (I) flow conditions for the two flow rate ranges considered, respectively. These tables also show the difference in mean error between the two flow regimes. Additionally, the table provides the average mean error and the corresponding standard deviation per meter type.

In the case of single-jet mechanical water meters there is a tendency towards over registration, significantly more pronounced at medium flows (200–500 L/h). This can also be observed in Figure 12 and Figure 13. However, in the same average flow rate range, the difference between steady and intermittent flow conditions for solid-state water meters reaches a maximum of +2.2% for M7 type meters. Compared to mechanical meters, this difference increases to +4.3% and +10.0% for M6 and M9 type meters, respectively. When conducting the same analysis in the upper flow rate range, from 700 to 2000 L/h, these differences are significantly reduced: for solid-state meters this parameter is always less than ±2.2% and for mechanical meters the maximum error difference is found for M9 meter type, reaching a value of +3.1%.

In summary, the error values obtained in the intermittent flow tests for solid-state water meters are significantly higher than those of single-jet mechanical water meters. However, on average, the behavior of the single-jet mechanical water meters is more deficient due to the appearance of positive bias, despite being more repetitive. In addition, solid-state water meters have frequently shown a null performance (errors reaching ± 100%) in the test under extreme conditions (cyclic period of 2 s) and the hypothesis that the errors may cancel out in the medium-long term cannot be rejected from the analysis conducted.

Figure 15 presents a series of boxplots charts that summarize the results obtained per meter type. In line with what has been previously stated, solid-state water meters show more dispersed behavior than single-jet mechanical water meters. Apart from the magnitude of this dispersion, it is important to note that the error distribution of the tests performed under intermittent flow includes the 0% error. Therefore, the exploitation of errors or any potential bias cannot be statistically confirmed with the amount of testing performed. Consequently, due to the lack of repeatability of the water meters, it would be necessary to test a larger sample and design a more detailed testing programme that includes specific assays to verify the accuracy of the meters under intermittent working conditions. In order to protect both the users and the water utilities, these test types will need to be included in the ISO 4064 and OIML R49 meter approval test programme, which should also keep records of the firmware version used by the meter.

## 4. Conclusions

The purpose of the research presented is to explore the stability of measuring errors of solid-state water meters subjected to intermittent and varying flow conditions. Performance under these working conditions is critical for water utilities as varying consumption flow rates are typical of domestic customers. However, these working conditions are not currently considered in any test programme defined in international standards related to water meters. 

The test programme proposed in this research is divided into two stages:(i)*Test under steady flow conditions*. The error of the meters is obtained by means of standing start and stop test method conducted in a volumetric test bench. These errors are taken as a reference for the results obtained in the next stage.(ii)*Test under intermittent flow conditions*. The designed programme includes different levels of both constant and variable flow rate, as well as different consumption durations.

The available sample consists of 28 solid-state water meters and seven mechanical meters, which are used as a reference. The resolution of the solid-state meters was not modified in the laboratory to ensure that the test conditions are as similar as possible to the real working conditions in the field.

The results obtained in tests under steady flow conditions show that all solid-state meters can maintain errors within the maximum permissible error of 2%. M7 type at Q_4_ is the exception, which showed an average error of approximately −8.5% at the highest flow rate tested of 5000 L/h. These results confirm the importance of testing the meters over a wide range of flow rates before they are put into operation.

On the other hand, it cannot be stated that the meters under analysis do not meet the repeatability requirements established in the ISO standard, provided that the reading resolution of the solid-state meters is limited to 1 L and, as a consequence, the overall uncertainty of the tests is close to 1%. For this reason, the ISO standard needs a significant improvement in relation to scale resolution requirements, which should be available without any interaction with the meters. Otherwise, it will be impossible to conduct accuracy tests in a laboratory that ensures that the meters under examinations will function during the tests exactly as they will in the field.

Regarding the variability of the error as a function of the flow rate, the error curve of solid-state water meters is relatively more uniform (flat) compared to the error curve of a single-jet mechanical water meter, which suffers from more oscillations throughout the measuring range. However, the ability of manufacturers to produce solid-state meters with the exact same performance does not show significant improvements in comparison with positive displacement meters. It was found that brand-new solid-state meters of the same brand significantly differ in terms of accuracy throughout the range (Figure 6 and Figure 7).

In contrast, the results obtained in tests under intermittent flow conditions show that the error magnitude significantly increases when compared to steady state conditions. It was not infrequent to obtain measuring errors of ±20% or greater. This difference in performance is affected by the cyclic periodicity of the flow, the duration of a consumption event being more significant than the flow rate of the test. In the case of mechanical meters, a clearer tendency towards over-registration as the duration of the consumption becomes shorter was observed. Conversely, solid-state water meters could suffer from a positive or negative bias due to intermittent flows. The ultrasonic meters tested were more influenced than the electromagnetic meters examined, mainly because the latter sample the flow signal more frequently and are, therefore, more adaptable for measuring short duration consumptions such as those found in households. Overall, it can be stated that the error dispersion of solid-state water meters has significantly increased. However, the probability distribution of the error differences between steady and intermittent flow conditions includes the 0% error. Therefore, exploitation of the errors or a potential bias cannot be statistically confirmed with the number of tests performed.

The results obtained suggest the need to design a more detailed testing programme that considers specific assays to verify the accuracy of the meters under intermittent working conditions, as well as the importance of including these types of tests in current water meter standards from ISO and OIML to protect users and water utilities.

## Figures and Tables

**Figure 1 sensors-20-05339-f001:**
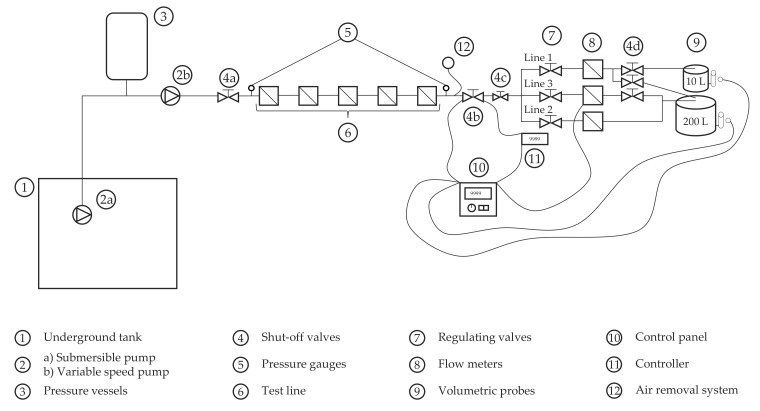
Test bench used to conduct tests under steady flow conditions.

**Figure 2 sensors-20-05339-f002:**
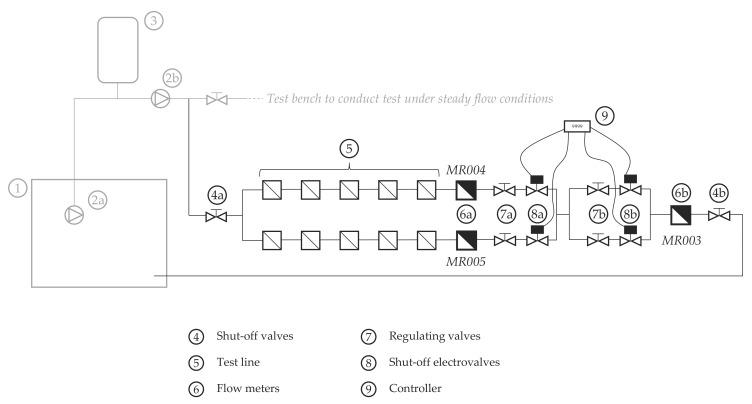
Test bench used to conduct tests under intermittent flow conditions.

**Figure 3 sensors-20-05339-f003:**
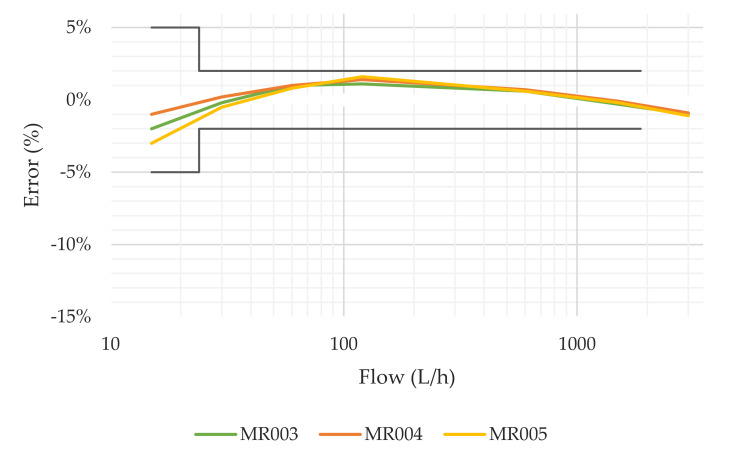
Error curves of the master meters installed in the test bench used for intermittent flows.

**Figure 4 sensors-20-05339-f004:**
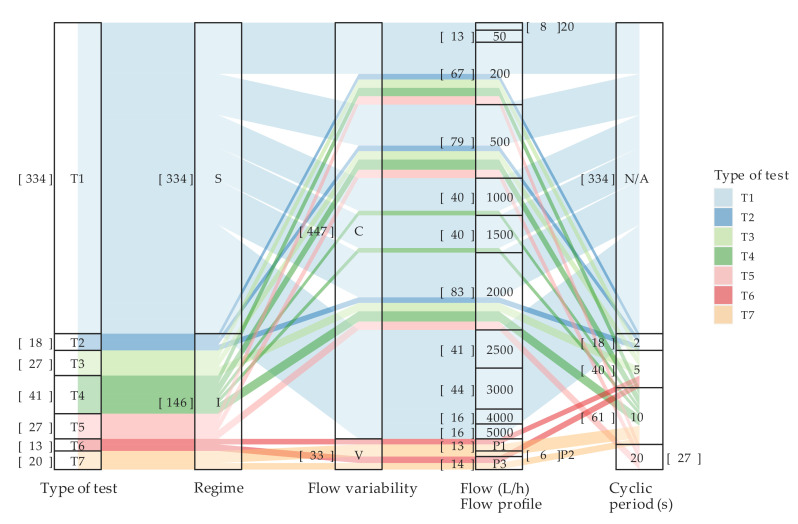
Distribution of the tests conducted according to the test type denomination, hydraulic regime (S = steady flow; I = intermittent flow), variability of flow (C = constant; V = variable), test flow rate or flow rate profile (P1, P2, or P3) and cyclic periodicity. The number in brackets indicates the quantity of tests conducted in each category.

**Figure 5 sensors-20-05339-f005:**
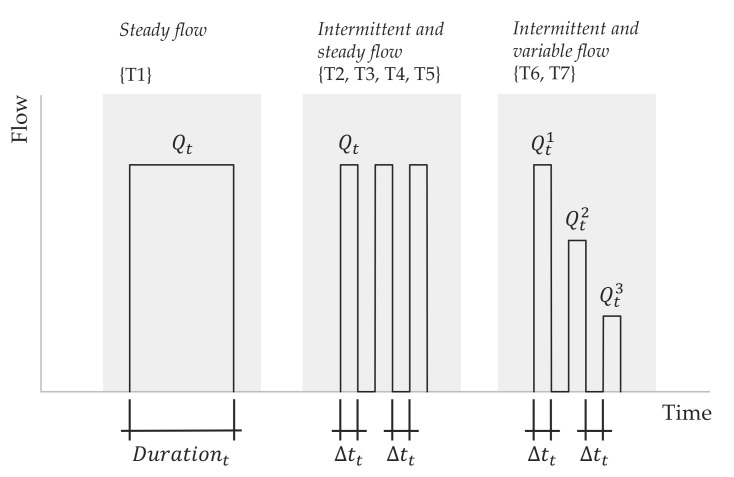
Parametrization of the tests conducted.

**Figure 6 sensors-20-05339-f006:**
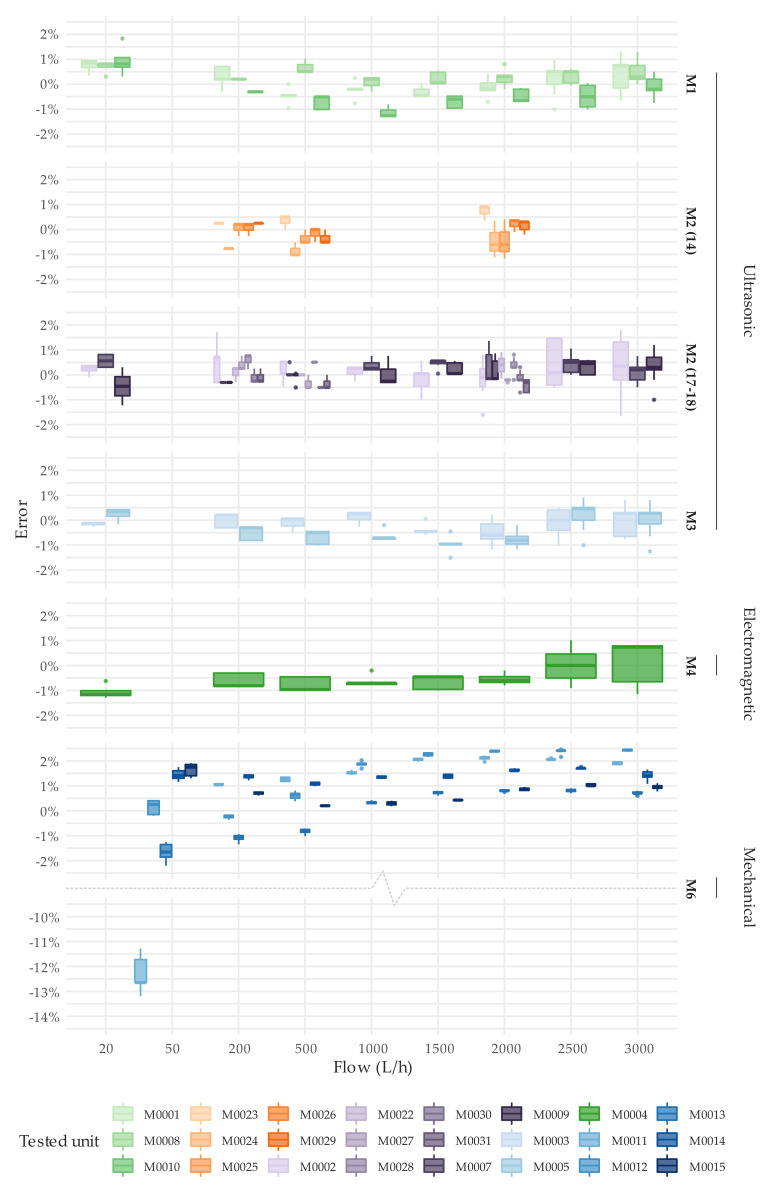
Error distribution under steady state flow conditions. Test type T1. DN15.

**Figure 7 sensors-20-05339-f007:**
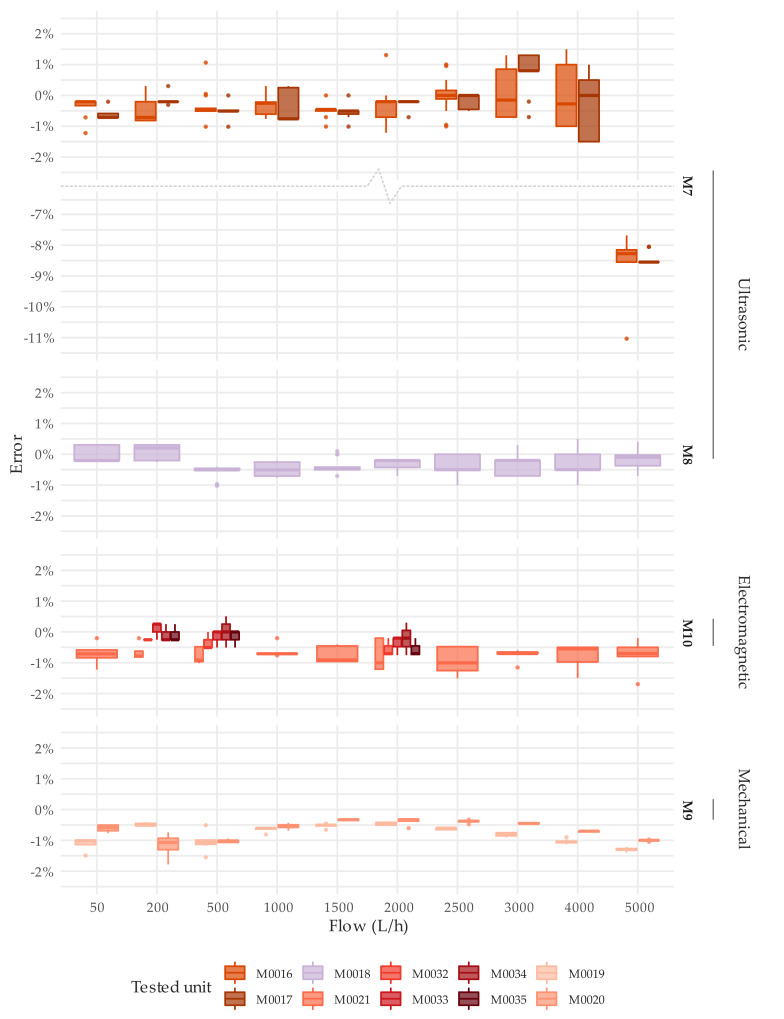
Error distribution under steady state flow conditions. Test type T1. DN20.

**Figure 8 sensors-20-05339-f008:**
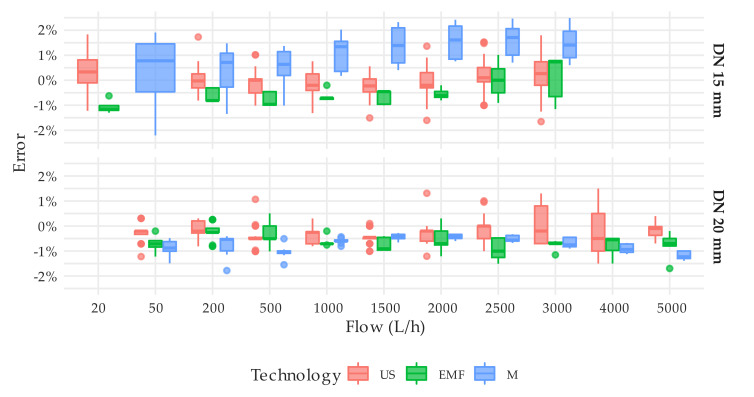
Variability of the standard deviation of the error obtained for tests under steady flow conditions.

**Figure 9 sensors-20-05339-f009:**
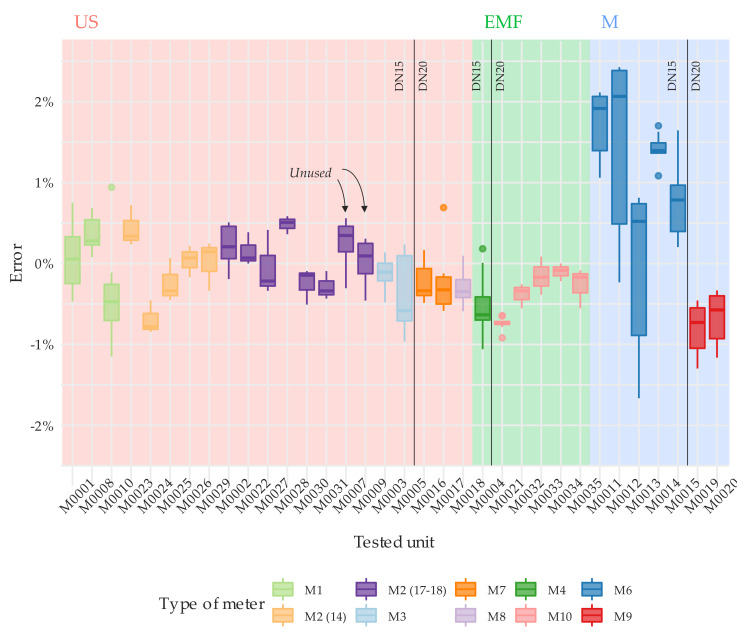
Error distribution throughout the tested flow rate range of the meters under analysis. T1 test type. Tests from M0011 (M6) at 50 L/h and from M0016 and M0017 (M7) at 5000 L/h were excluded in order not to bias the results.

**Figure 10 sensors-20-05339-f010:**
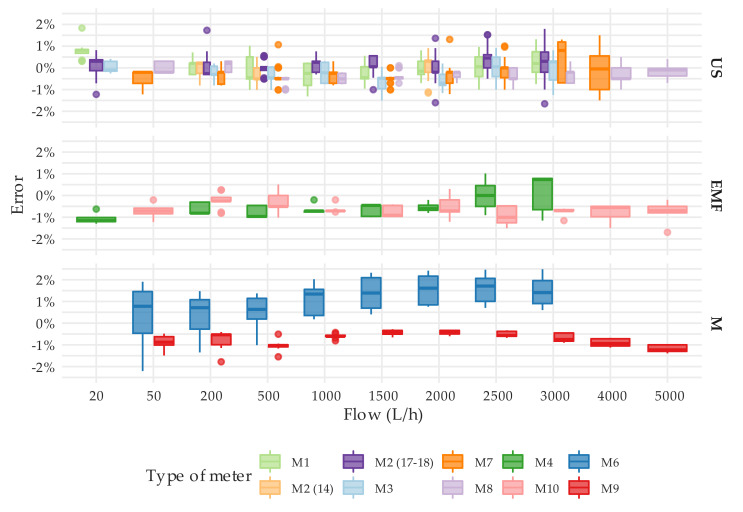
Error distribution by flow rate, technology and meter type. T1 test type. Tests on the M0011 (M6 meter type) at 50 L/h and on the M0016 and M0017 (M7 meter type) at 5000 L/h were excluded in order not to bias the results.

**Figure 11 sensors-20-05339-f011:**
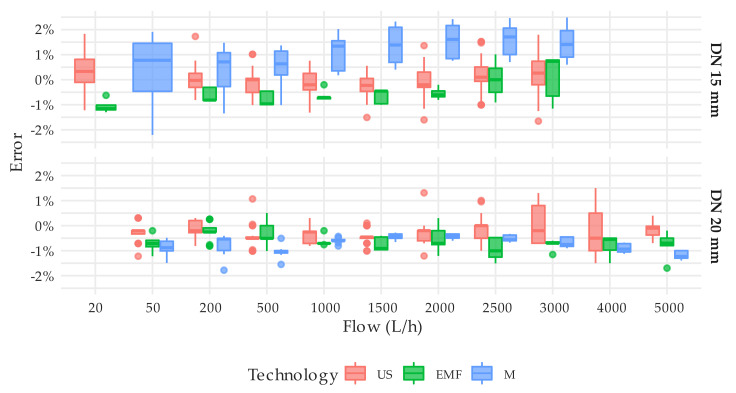
Error distribution by flow rate, diameter, and technology. T1 test type. Tests on the M0011 (M6 meter type) at 50 L/h and on the M0016 and M0017 (M7 meter type) at 5000 L/h were excluded in order not to bias the results.

**Figure 12 sensors-20-05339-f012:**
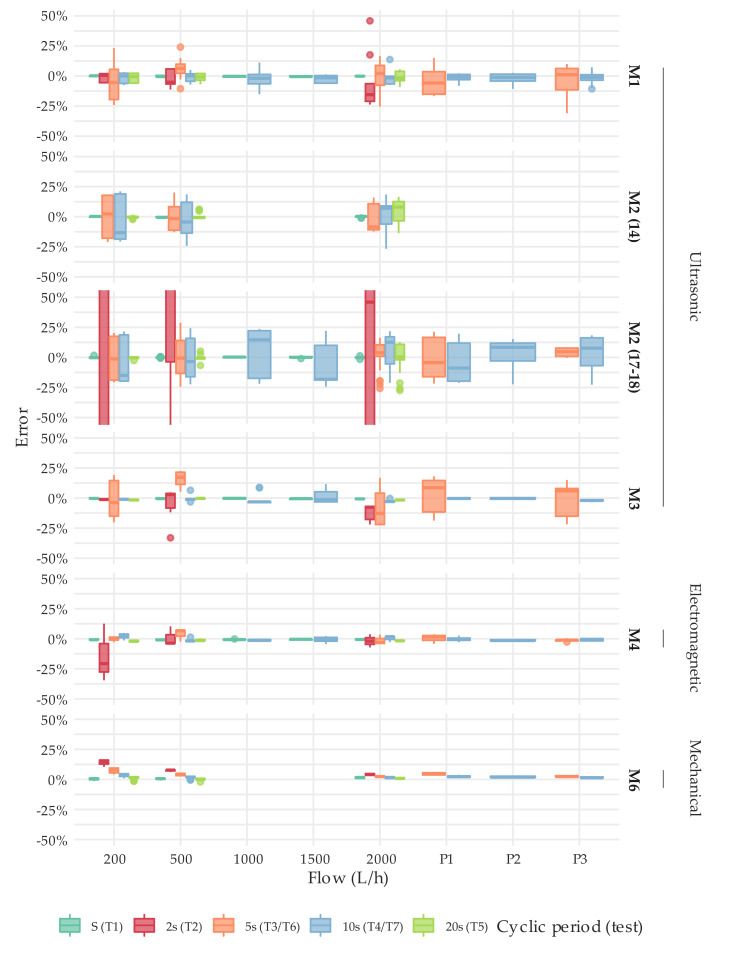
Error distribution under intermittent flow conditions. T2, T3, T4, T5, T6, and T7 test types. DN15. Results of T1 type tests were added as a reference.

**Figure 13 sensors-20-05339-f013:**
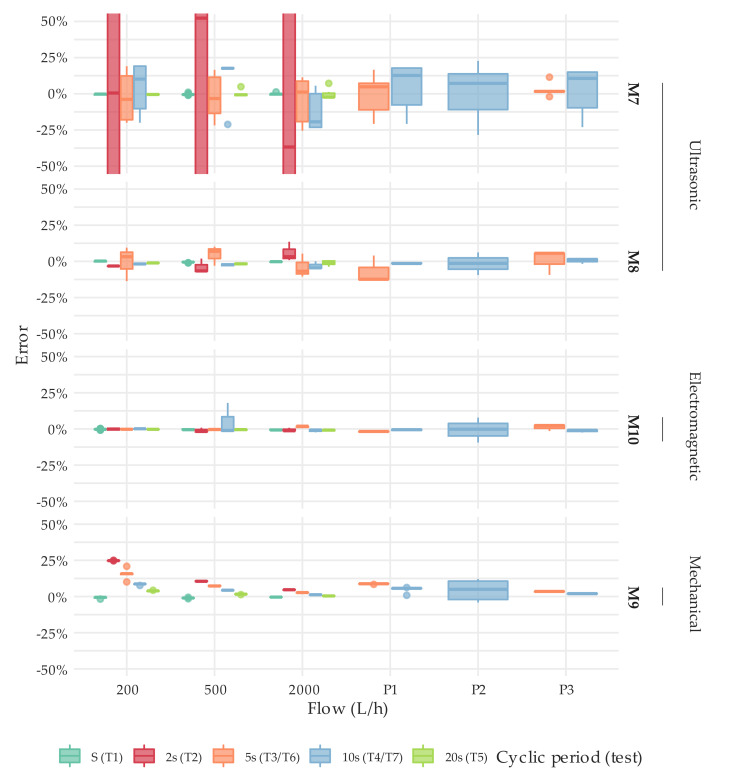
Error distribution under intermittent flow conditions. T2, T3, T4, T5, T6, and T7 test types. DN20. Results of T1 type tests were added as a reference.

**Figure 14 sensors-20-05339-f014:**
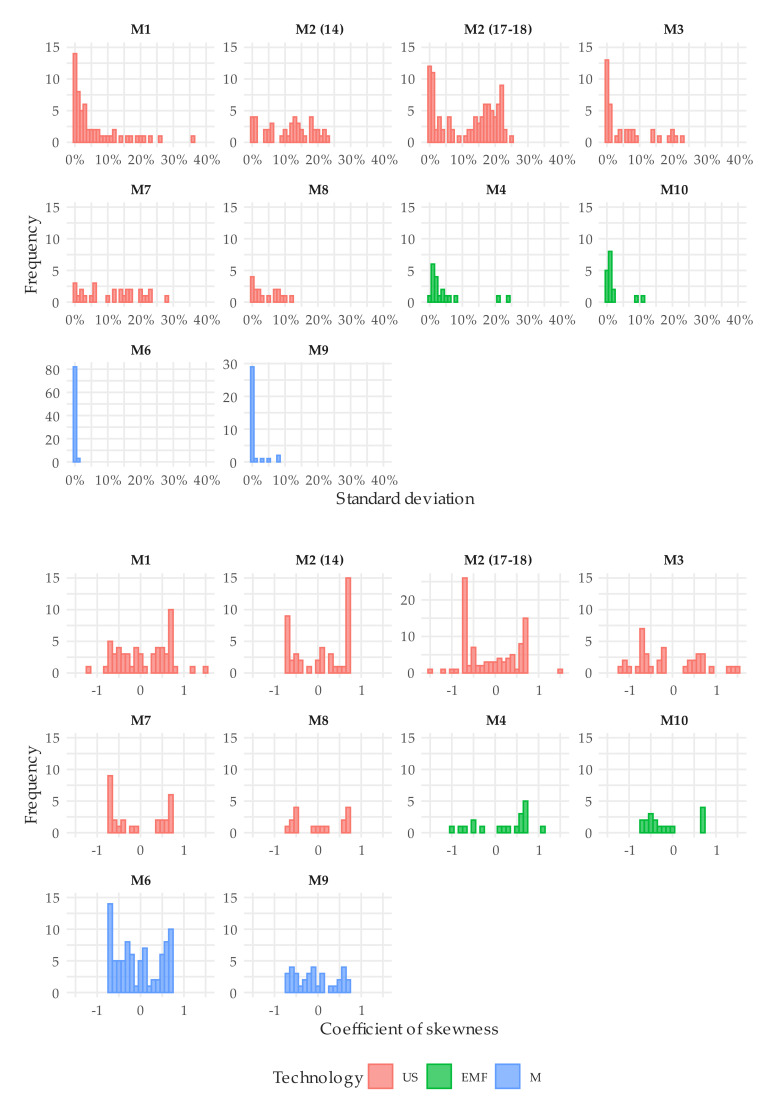
Histogram of the standard deviation and coefficient of skewness corresponding to error distribution under intermittent flow conditions (Figure 12 and Figure 13). The color of the bars depends on the type of technology. The bar width is 0.1% and 0.1, respectively.

**Figure 15 sensors-20-05339-f015:**
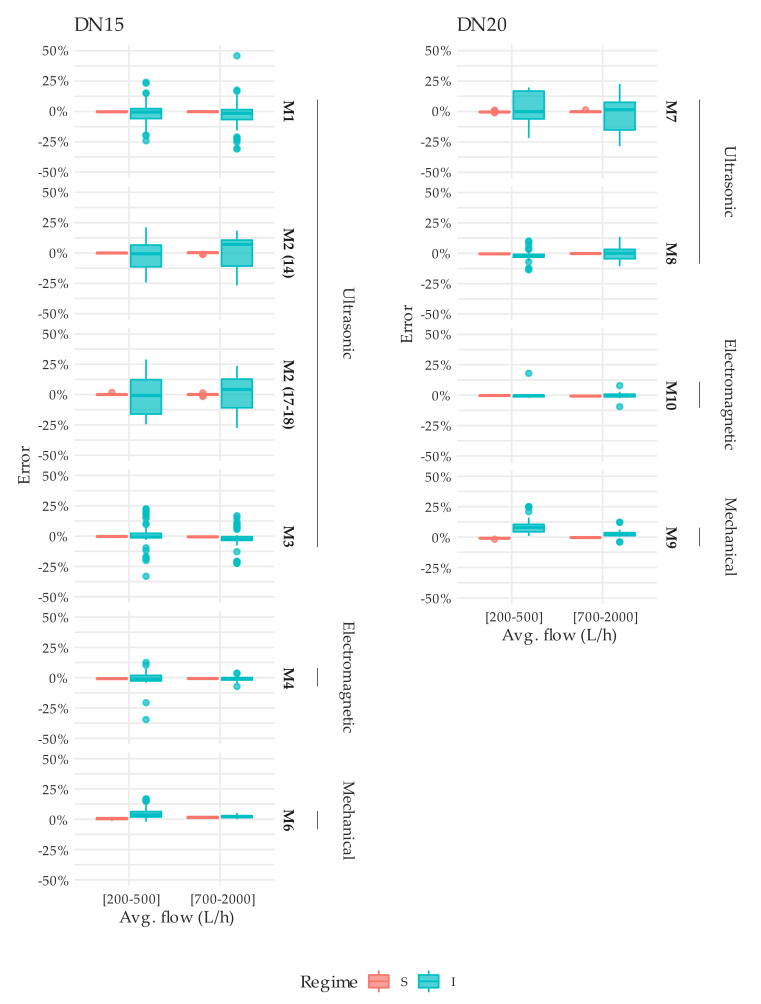
Distribution of errors obtained in the tests under steady (S) and intermittent (I) flow conditions at an average flow rate of 200–500 and 700–2000 L/h.

**Table 1 sensors-20-05339-t001:** Tested meters distributed by manufacturer, type, technology (EMF = electromagnetic, US = ultrasonic, M = mechanical), meter size, and metrological class. (*) The meter type M2 is divided in two subcategories according to their age: M2 (14) are meters manufactured in 2014 and M2 (17–18) are meters manufactured in 2017–2018.

Manufacturer	Type of Meter	Num. of Units	Technology	DN	Metrological Class (Q_3_/Q_1_)	Q_1_ (L/h)	Q_2_ (L/h)	Q_3_ (m^3^/h)	Q_4_ (m^3^/h)
B3	M1	3	US	15	400	6.25	10	2.5	3.125
B5	M2 (14)^(*)^	5	US	15	160	10	16	1.6	2.0
B5	M2 (17–18)^(*)^	8	US	15	160	10	16	1.6	2.0
B1	M3	2	US	15	400	6.25	10	2.5	3.125
B4	M4	1	EMF	15	800	3.125	5	2.5	3.125
B7	M5	1	US	15	800	3.125	5	2.5	3.125
B2	M6	5	M	15	125	20	32	2.5	3.125
B5	M7	2	US	20	250	10	16	2.5	3.125
B6	M8	1	US	20	400	10	16	4.0	5.0
B4	M9	2	M	20	160	25	40	4.0	5.0
B4	M10	5	EMF	20	800	5	8	4.0	5.0

**Table 2 sensors-20-05339-t002:** Average age and registered volume of the meters as received in the laboratory. Additionally, the default resolution and the availability of resolution change per meter type.

Type of Meter	Age (Years)	Accumulated Volume (m^3^)	Default Resolution	Resolution Change
M1	1	3.8	L	Available
M2	3.6	2611.8	L	Available
M3	1.0	1.4	L	Available
M4	5.0	0.2	L	Available
M5	2.0	12.7	L	Available
M6	4	2289.5	dL	Not available
M7	0.5	1.2	L	Available
M8	0	0.1	L	Available
M9	0	0.7	dL	Not available
M10	0	0.4	L	Available

**Table 3 sensors-20-05339-t003:** Duration, average volume, and reading uncertainty per test type.

Type of Test	Avg. Duration (min)	Avg. Volume (L)	Reading Uncertainty
T1	32	195	1.02%
T2	80	504	0.40%
T3	72	396	0.50%
T4	69	486	0.41%
T5	73	388	0.52%
T6	98	790	0.25%
T7	106	734	0.27%

**Table 4 sensors-20-05339-t004:** List of meters that present abnormal performance (errors above 50% of reference volume) in the specified tests.

Technology	Type of Meter	ID Meter	Test	Time Frame (s)
US	M2	M0002	T2	2
		M0007	T2	2
		M0009	T2	2
	M7	M0016	T2	2
		M0017	T2	2

**Table 5 sensors-20-05339-t005:** Mean and standard deviation of errors obtained in tests under steady (S) and intermittent (I) flow conditions at an average flow rate of 200–500 L/h. It also includes the mean difference between the two regimes.

Tech.	Diam.	Type of Meter	Meter	${\bar{x}}_{\left[200-500\right]}^{S}$	$s_{\left[200-500\right]}^{S}$	${\bar{x}}_{\left[200-500\right]}^{I}$	$s_{\left[200-500\right]}^{I}$	${\bar{x}}^{S}-{\bar{x}}^{I}{}_{\left[200-500\right]}$
US	DN15	M1	M0001	−148%	0.518%	1.952%	8.202%	2.100%
			M0008	0.470%	0.296%	−1.957%	7.171%	−2.426%
			M0010	−0.546%	0.296%	−4.145%	7.347%	−3.599%
			***Avg. M1***	−0.075%	0.512%	−1.383%	3.088%	−1.308%
		M2 (14)	M0002	0.318%	0.609%	5.103%	16.020%	4.785%
			M0007	−0.081%	0.243%	−0.834%	14.756%	−0.753%
			M0009	−0.165%	0.190%	−3.597%	13.740%	−3.432%
			M0022	0.034%	0.208%	−1.962%	12.691%	−1.995%
			M0027	0.039%	0.491%	0.841%	14.658%	0.803%
			M0028	0.546%	0.200%	2.059%	13.504%	1.512%
			M0030	−0.300%	0.296%	−2.871%	13.640%	−2.571%
			M0031	−0.216%	0.298%	−6.307%	12.172%	−6.091%
			***Avg. M2 (14)***	0.022%	0.285%	−0.946%	3.572%	−0.968%
		M2 (17–18)	M0023	0.288%	0.194%	−2.206%	13.511%	−2.494%
			M0024	−0.813%	0.190%	2.480%	11.103%	3.293%
			M0025	−0.135%	0.340%	−2.151%	10.559%	−2.016%
			M0026	−0.051%	0.287%	−0.252%	12.483%	−0.201%
			M0029	−0.046%	0.370%	−2.430%	12.721%	−2.383%
			***Avg. M2 (17–18)***	−0.040%	0.318%	−0.934%	2.866%	−0.894%
		M3	M0003	−0.063%	0.259%	1.466%	9.085%	1.529%
			M0005	−0.614%	0.279%	−0.267%	10.202%	0.346%
			***Avg. M3***	−0.339%	0.389%	0.599%	1.226%	0.938%
	DN20	M7	M0016	−0.389%	0.443%	0.067%	13.139%	0.456%
			M0017	−0.343%	0.326%	3.635%	14.761%	3.978%
			***Avg. M7***	−0.366%	0.033%	1.851%	2.523%	2.217%
		M8	M0018	−0.322%	0.410%	−1.800%	5.430%	−1.478%
EM	DN15	M4	M0004	−0.699%	0.281%	−1.113%	8.121%	−0.415%
	DN20	M10	M0021	−0.716%	0.267%	−0.004%	3.498%	0.712%
			M0032	−0.296%	0.191%	-	-	-
			M0033	−0.042%	0.297%	-	-	-
			M0034	−0.042%	0.374%	-	-	-
			M0035	−0.127%	0.267%	-	-	-
			***Avg. M10***	−0.245%	0.283%	−0.004%	-	0.241%
M	DN15	M6	M0011	1.169%	0.127%	3.873%	2.773%	2.705%
			M0012	0.218%	0.452%	2.916%	4.242%	2.698%

**Table 6 sensors-20-05339-t006:** Mean and standard deviation of errors obtained in tests under steady (S) and intermittent (I) flow conditions at an average flow rate of 700–2000 L/h. It also includes the mean difference between the two regimes.

Tech.	Diam.	Type of Meter	Meter	${\bar{x}}_{\left[700-2000\right]}^{S}$	$s_{\left[700-2000\right]}^{S}$	${\bar{x}}_{\left[700-2000\right]}^{I}$	$s_{\left[700-2000\right]}^{I}$	${\bar{x}}^{S}-{\bar{x}}^{I}{}_{\left[700-2000\right]}$
US	DN15	M1	M0001	−0.221%	0.300%	2.914%	10.575%	3.134%
			M0008	0.190%	0.286%	−1.886%	6.899%	−2.076%
			M0010	−0.773%	0.374%	−8.652%	8.970%	−7.879%
			***Avg. M1***	*−0.268%*	*0.483%*	*−2.541%*	*5.810%*	*−2.273%*
		M2 (14)	M0002	−0.076%	0.549%	6.451%	12.804%	6.527%
			M0007	0.383%	0.400%	−0.614%	15.668%	−0.997%
			M0009	0.142%	0.387%	2.262%	15.288%	2.121%
			M0022	0.387%	0.530%	1.460%	10.937%	1.073%
			M0027	−0.217%	0.038%	−2.309%	16.240%	−2.092%
			M0028	0.362%	0.330%	−0.049%	16.749%	−0.410%
			M0030	−0.145%	0.330%	0.684%	14.733%	0.828%
			M0031	−0.434%	0.260%	2.455%	14.996%	2.889%
			***Avg. M2 (14)***	*0.050%*	*0.314%*	*1.292%*	*2.611%*	*1.242%*
		M2 (17–18)	M0023	0.723%	0.321%	−2.256%	9.746%	−2.979%
			M0024	−0.454%	0.743%	0.610%	10.563%	1.064%
			M0025	−0.454%	0.793%	4.681%	9.557%	5.135%
			M0026	0.219%	0.278%	2.807%	15.606%	2.588%
			M0029	0.145%	0.240%	0.899%	14.684%	0.754%
			***Avg.M2 (17–18)***	*0.045%*	*0.360%*	*1.312%*	*2.397%*	*1.267%*
		M3	M0003	−0.244%	0.418%	−3.924%	8.999%	−3.679%
			M0005	−0.798%	0.297%	−1.357%	7.395%	−0.559%
			***Avg. M3***	*−0.521%*	*0.391%*	*−2.640%*	*1.815%*	*−2.119%*
	DN20	M7	M0016	−0.295%	0.571%	1.753%	10.762%	2.048%
			M0017	−0.274%	0.191%	−5.312%	16.424%	−5.039%
			***Avg. M7***	*−0.284%*	*0.015%*	*−1.780%*	*4.996%*	*−1.495%*
		M8	M0018	−0.331%	0.218%	−0.372%	6.026%	−0.041%

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
