# Peer review of "Accuracy of Solid-State Residential Water Meters under Intermittent Flow Conditions"

_sensors, 2020, doi:10.3390/s20185339_

Round 1

Reviewer 1 Report

The accuracy of water meters in residential customers is critical to both the customers and water utilities. The sampling frequency of solid-state meters including ultrasonic meter and electromagnetic meter has a significant effect on the accuracy of the instruments when measuring intermittent flow. A lot of research has been done to investigate the metrological performance of solid-state water meters in the presence of intermittent flows with different durations. However how to solve this problem is not answered in technical perspective. Why the accuracy of ultrasonic meter is lower than electromagnetic meter can be discussed though the accuracy of meter is determined by the specific type of meter. The experimental results take up too much space. The paper can be organized in a better manner.

Reviewer 2 Report

The Authors report the experimental results of several lab tests in order to examine the accuracy of solid-state water meters in residential field. This represent a very interesting topic. They compared three class of meters: ultrasonic, electromagnetic and mechanical, both in steady state and in intermittent flow conditions. In addition, authors investigate the impact of sampling frequency and the effects of firmware on accuracy of smart meters.

The paper is well structured. The introduction and methodology sections are well described. Authors provide enough information about test benches used to conduct experiments in steady and intermittent flow conditions in that way experiments can be reproduced easily.

The obtained results are sufficiently discussed and the data reported in supplementary materials are very useful to better understanding the study. The conclusions are adequate. Nevertheless, some minor revisions are required to improve the manuscript.

Minor comments

  • Could Authors specify the values of Q4 (lines 72), Q1 (line 144) and Q2 (line 401)?
  • Lines: 385-386: “This malfunctioning at high flows has also been detected by the authors in other brands of ultrasonic meters”, please provide a reference.
  • Lines: 387-388: “In any case, this is clearly related to the firmware of the meter rather than an evident malfunctioning of any component.” It is not easily discerning the previous statement from the reported results. Could authors provide more details?
  • Lines: 432-433: “Unfortunately, once the test mode is activated there is no means of guaranteeing that the meter will have the exact same performance as with the test mode deactivated.” Have authors proof of different behaviour between normal and test mode?
  • Lines: 542-543: “Thus, the large dispersion of the errors found is strictly related to the internal algorithms and signal sampling periodicity.” Based on reported results, how is it possible relate the errors to internal algorithms?
  • Figure 15: please report some comments about Coefficient of skewness.
  • Liens 488: please adding some recently published papers on residential water end-use such as:

Cominola, A., Nguyen, K., Giuliani, M., Stewart, R. A., Maier, H. R., & Castelletti, A. (2019). Data mining to uncover heterogeneous water use behaviors from smart meter data. Water Resources Research, 55, 9315– 9333. https://doi.org/10.1029/2019WR024897

Di Mauro, A.; Di Nardo, A.; Santonastaso, G.F.; Venticinque, S. Development of an IoT System for the Generation of a Database of Residential Water End-Use Consumption Time Series. Environ. Sci. Proc. 2020, 2, 20. ttps://doi.org/10.3390/environsciproc2020002020

Reviewer 3 Report

A very good piece of research and a well-written manuscript.

  • Minor editing/proof-reading required. For instance line 15, instead of asall should be "as all". Check line 46 "measuring large customers" - revise because you are not actually measuring customer, you are measuring water consumption.Revise the use of the word "essays" in line 299 and 680 - it does not appear to be correct.
  • Please highlight the major findings of your research in the Abstract.
  • Figures 4 and 5 are a little confusing. I suggest you improve their presentation. Not clear what M2(17-18) means. The colour cording in Figure 5 is not explained. You may wish to convert to tables so that they can be easily understood.

Round 2

Reviewer 1 Report

What is the proportion of water consumption events of 30 s or less to the total evens in residential customers according to reference 14-17? What about percentage of the total water consumption of this part accounts for the total water consumption? These are important indicators to measure the importance of the issue, and they are also important references for ISO to formulate the standard. Higher requirements also mean higher costs, which is usually paid by the government or water supply unit.

The water supply unit may be more interested in the conclusions of the paper, because this is an issue related to the fairness, which can also be seen from the references in the paper. Designers of sensors or meters may be more interested in how to solve this problem, such as low-power design techniques, high energy-density batteries, or replaceable volumetric flow meters that are widely used in many countries.

Author Response

Please find answer in the attached document
